Journal of Data-centric Machine Learning Research (2026)          Submitted 8/25; Revised 1/26; Published 4/26

# Time Series Machine Learning for Classifying Electroencephalograms

**Aiden Rushbrooke**                                          AIDEN.RUSHBROOKE@UEA.AC.UK
*School of Computing Sciences*
*University of East Anglia*
*Norwich, UK*

**Saber Sami**                                                     S.SAMI@UEA.AC.UK
*Norwich Medical School*
*University of East Anglia*
*Norwich, UK*

**Matthew Middlehurst**                          M.MIDDLEHURST@BRADFORD.AC.UK
*School of Computing and Engineering*
*University of Bradford*
*Bradford, UK*

**Anthony Bagnall**                                   A.J.BAGNALL@SOTON.AC.UK
*School of Electronics and Computer Science*
*University of Southampton*
*Southampton, UK*

**Reviewed on OpenReview:** *https: // openreview. net/ forum? id= oPQpQlsfDO*

**Editor:** Fernando Perez-Cruz

## Abstract

Electroencephalography (EEG) is a crucial tool across neuroscience domains, including medical diagnostics, psychological research, and brain-computer interfacing (BCI). Its popularity is due to its non-invasiveness, high temporal resolution, and cost-effectiveness. The task of EEG classification involves learning to predict class labels associated with EEG segments based on previously observed data. This task is fundamental yet complex, given the high dimensionality, variability, and subject-specific nuances inherent in EEG data. We systematically evaluate recent advances in general-purpose time series machine learning (TSML) approaches to EEG classification. We present an EEG classification archive of 30 benchmark datasets, spanning diverse applications from clinical diagnostics to cognitive and BCI tasks. Our empirical evaluation compares traditional EEG approaches, deep learning models, Riemannian geometry-based classifiers, and state-of-the-art time series machine learning algorithms on this new benchmark. We find that one algorithm, a meta-ensemble called HIVE-COTE v2.0, consistently outperforms alternative classifiers.

## 1 Introduction

Electroencephalography (EEG) is a commonly used non-invasive technique for recording neural activity. It utilises electrodes placed on the scalp to monitor the electrical signals produced by neurons and records them as numerical values. EEG is popular for its ease of

recording, low cost and high temporal resolution, and has become a staple in many different neuroscientific fields such as medicine (Bera, 2021), psychology (Yun, 2024) and brain-computer interface (BCI) research (Wolpaw et al., 2002). Among EEG's many applications, we focus here on classification. The task is to learn a model to predict, for example, whether a subject is moving their arm or has a medical condition. This may be the ultimate goal, such as in the development of a neural controller for a device, or part of a broader investigation into biomarkers.

Specifying an EEG classification problem requires substantial preprocessing and a number of design decisions. We characterise the process of learning a classification model from EEG in three phases: preprocessing; transformation or feature creation; and model learning. The first two phases are the focus of most EEG research and constitute a central part of the problem specification.

In the preprocessing phase, the objective is to convert raw EEG recordings (often annotated with labels and collected across multiple subjects and sessions) into a structured classification dataset. This involves preparing well-defined cases (i.e., EEG recordings) paired with corresponding labels, and organising them into training and testing sets suitable for model development and evaluation. For example, an EEG experiment may be designed to see if we can predict if a subject is looking at a picture of a human face or random noise[1]. The EEG of a subject is recorded during a session where they could, for example, be shown an image every five seconds. A single recording session is then preprocessed through, e.g. removal of artefacts not associated with the classification task (such as blinking or periods of inactivity between events) and/or filtering to limited bandwidths. The data may be further transformed through feature creation. For example, the time domain signal might be transformed into the time-frequency domain or summarised into power bands. The data are then usually segmented into epochs each associated with a class label, with which it forms a single case in the classification problem. Further transformation may then be applied per epoch/case.

In the majority of the literature, classification of EEG follows one of two forms. Traditionally, descriptive features are derived over each epoch and a standard classifier such as LDA or random forest is applied (Steyrl et al., 2014). More recently, deep learning models that learn directly from EEG have been proposed, with Convolutional Neural Networks (CNNs) being the most popular (Hossain et al., 2023).

An unexplored alternative is to use time series machine learning (TSML) algorithms for EEG classification. The field of TSML, and more specifically time series classification (TSC), proposes general purpose estimators specifically for ordered data series (Middlehurst et al., 2024b) and evaluates them on a broad range of data archives (Dau et al., 2019). Our aim is to assess how well these general purpose TSC algorithms perform on EEG classification problems and to compare to existing EEG classification methods. The multivariate TSC archive (Ruiz et al., 2021) includes six EEG tasks. We have extended it to create an EEG classification archive of 30 problems with pre-defined train/test splits. We focus primarily on the TSC algorithm HIVE-COTE v2.0 (HC2) (Middlehurst et al., 2021). HC2 is a meta-ensemble of classifiers that are built on different kinds of discriminatory features. We experimentally demonstrate that HC2 is the best performing classifier on our EEG clas-

---

1. `https://www.kaggle.com/c/decoding-the-human-brain/data`

sification archive. We highlight its strengths and weaknesses, before proposing ways that HC2 could be better adapted for high dimensional data such as EEG.

The rest of this paper is structured as follows. In Section 2 we describe the EEG archive and discuss formatting issues. In Section 3 we provide detail on the classifiers used in experimentation. In Section 4 we present results. In Section 5 we investigate the factors contributing to the performance of HC2 and in Section 6 we describe our conclusions and future directions.

## 2 The EEG Classification Archive

Our goal is to identify open source EEG classification datasets to form the basis of the comparison of algorithms. We assume any required pre-processing such as artefact removal and the specification of epochs has been conducted by the data owners and we have arrived at the point where we have a collection of cases, where each case is a multivariate EEG reading (possibly of varying duration) and an associated class label. Achieving this is non-trivial, and may present several complicating factors. A core assumption of classification problem specification is that cases are independent. EEG classification nearly always violates this assumption through the preprocessing and transformation stages. For example, multiple cases are usually recordings from a single subject and from the same recording session. Artefact removal may not be easily generalisable to unseen cases. Transformation may use information that spans a whole session, making it hard to generalise to a single unseen case. We view the transformation and/or feature creation of EEG as part of the classification pipeline. Transformation that relies on data characteristics that span epochs must be conducted independently on the train and test sets. It is also not always clear whether the classification model is meant to be independent of the subjects used in experimentation. These factors mean care needs to be taken when evaluating models, since all of these factors may introduce bias into model evaluation.

Our preference is for data that has been used in peer-reviewed publications that has classification experiments we can potentially reproduce. There is a huge volume of open source EEG available. However, only a small number of these datasets are used for classification, and those that are typically rely on bespoke formats and data storage conventions. Most are separated by trial and require significant preprocessing, with the design decisions that entails, in order to formulate it as a classification problem. We aim to select datasets from a variety of different EEG fields, including BCI, medical, and psychology. We also aim to include a range of different EEG problem types, from both real and imagined tasks as well as active and resting state data.

Some datasets come from larger EEG experiments with many different tasks and potential classification problems.

In these we evaluate on a case-by-case basis in order to avoid biases toward any single dataset by either selecting a single problem from that dataset, or in cases where we believe the problems are distinct and small enough we separate into different problems. We also indicate where this has occurred in our dataset explanations.

We have collated and formatted 30 problems. We take the data as it is presented, conducting no preprocessing of our own. We primarily aimed to collect only raw data wherever available in order to provide a uniform archive. As processing is usually a manual

task, often involving expert knowledge to detect artefacts and dataset specific techniques, we decided to avoid applying any preprocessing steps ourselves. In the few cases where the raw data is not available, we have reported any preprocessing steps that were described by the original authors. Table 1 lists the datasets. Most datasets have 32, 64 or 128 channels, although some were collected with specific equipment that utilises different numbers of channels, and some have been altered to remove faulty or non-EEG channels.

We have created a default train/test split to simplify reproducibility. For datasets containing multiple participants, we ensure that no recordings of a single participant are found in both splits to avoid biases caused by the classifier detecting recording sessions rather than useful information. For single participant datasets we split based on recording session. In cases with multiple participants we also provide an ancillary file containing the subject index for each instance.

Six of the datasets in our new EEG archive are part of the existing MTSC archive (Ruiz et al., 2021). Five of these (FingerMovements, HandMovementDirection, MotorImagery, SelfRegulationSCP1 and SelfRegulationSCP2) are taken from the BCI competitions and have all been preprocessed and further information can be found at the BBCI website[2]. FaceDetection is from a Kaggle competition[3]. Of the remaining 24, eight are related to medical conditions and 16 are BCI related. A majority of the datasets are uploaded and available on Zenodo[4], and the remaining will be made available soon. The datasets can be downloaded directly from Zenodo or in code using `aeon`. Code to download and reproduce results can be found on the `aeon-neuro` GitHub repository in the benchmarking section[5].

### 2.1 Medical Classification

One characteristic of medical EEG studies is that the cost and complexity of collecting data is high: recruiting patients is difficult and ethics policies must be rigorously followed. Studies containing 100 subjects are considered large. The lack of data means studies are usually designed to investigate specific hypotheses about biomarkers or the effect of a condition on the ability of a subject to perform a task, rather than attempt to develop diagnostic tools. Nevertheless, the potential benefits of early diagnosis of conditions such as Alzheimer's means that the investigation of predictive properties of EEG based classifiers is always a useful secondary analysis.

**Alzheimers** (Miltiadous et al., 2023) aims to predict if a subject has Alzheimer's (36 individuals), frontotemporal dementia (23), or is a healthy control subject (29). Labelling is based on Mini-Mental State Examination scores taken over a period of time. EEG was recorded in eyes closed resting state. We extract a 60 second window of resting state data for each subject, recorded at 500Hz with 19 EEG channels.

**PhotoStimulation** (Ntetska et al., 2025) is a dataset similar to **Alzheimers**, with the same classification problem and the data collected from the same cohort, but at a later time. Each participant was placed in front of a screen with their eyes open and exposed to a photic stimulation at varying frequencies, starting at 5Hz and increasing by 5Hz up

---

2. `https://bbci.de/competition/`

3. `https://www.kaggle.com/c/decoding-the-human-brain/data`

4. `https://zenodo.org/communities/tsml/records?q=metadata.subjects.subject%3A%22eeg%22&l=list&p=3&s=10&sort=bestmatch`

5. `https://github.com/aeon-toolkit/aeon-neuro`

to 30Hz. We select an 18 second window from the 10Hz range for our dataset as that was found to contain the most consistent data across subjects. We removed 15 participants due to data irregularities, leaving 73: 31 with Alzheimer's, 22 with frontotemporal dementia, and 20 controls. The data was recorded at 500Hz with 19 EEG channels.

**EpilepticSeizures** (Andrzejak et al., 2001) is a single channel EEG study looking into epileptic seizures. For each subject, 23.6 seconds of brain activity was monitored, and segmented into 1 second windows. The data was collected at 178Hz and split into train and test partitions, with the train set containing 80 of the 11500 instances. Originally, the dataset was split into 5 classes relating to eye state, tumours, and seizures. The original data was subsequently formatted into the two class problem by merging classes, where the goal is predicting if the associated subject is experiencing a seizure or not (Zhang et al., 2022).

**ShortIntervalTask** and **LongIntervalTask** (Singh et al., 2023) look at the impact of Parkinson's disease on completing a basic task. 94 individuals with Parkinson's, along with 45 control participants, were asked to complete a simple task where they had to hold down a button for either 3 or 7 seconds, self timed, based on a visual cue. This was repeated a number of times for each participant, swapping randomly between the short and long intervals.

The data was recorded at 500Hz with 63 EEG channels. From this two classification problems were formed for the task discriminating between the participants with Parkinson's and the control for both the short and long interval tasks.

**FibroUEA** and **FibroLiverpool** datasets are both part of larger studies into Fibromyalgia Syndrome (FMS). The problem we address is diagnosis: classifying whether a subject suffers from FMS or not. The University of East Anglia study (FibroUEA) was part of a wider project based around the use of virtual reality to treat participants with FMS. The dataset consists of 27 individuals with FMS, and 14 healthy controls, and was recorded with 64 EEG channels at 500Hz. Each session lasted slightly under 2 minutes. Sessions were truncated to the smallest so all were equal lengths. A similar trial with different subjects gave rise to the FibroLiverpool data, recorded at the University of Liverpool. This experiment has 19 individuals with FMS, and 18 healthy controls. FibroLiverpool is eyes open resting state data recorded with 64 channels at 512Hz.

### 2.2 Brain Computer Interface Classification

BCI tasks are nearly always classification problems involving detecting movement based on EEG. One common distinction is between detecting real or imagined movement.

**Blink** (Chicaiza and Benalcázar, 2021) aims to differentiate between short and long blinks. Each of the subjects was asked to blink for a duration of two seconds, as either a short or long blink. Each case is two seconds sampled at 255Hz (510 observations), with 4 EEG channels. The data were collected in 20 trials of six subjects, with 50 blinks per experiment. One of the long blink data files in the original repo was a duplicate, so we have removed this.

The first five long and short blink trials are used for the train data (250 long blink, 250 short blink). The remaining 200 long blink and 250 short blink constitute the test data set.

**ButtonPress** is a very basic dataset composed of EEG recordings of a single participant either pressing a button or idle, collected as part of a training exercise for EEG use. The data was recorded with 32 channels at 1000Hz, and the subject completed 120 button presses in total, with each instance composed of 1 second around the press.

**EyesOpenShut** (Roesler and Suendermann, 2013) aims to predict if a subject's eyes are open or shut. The data is recorded on a single patient for 117 seconds with a 14 channel EEG at 128Hz. The original formulation treated each observation (1/128th of a second) as a case. We remove obvious outliers (reading less than 3000 or more than 5000), segment the data into 1 second intervals and retain only the intervals where the eyes are either open (class 0) or shut (class 1). This gives a 14 channel problem with 128 observations per dimension.

**OpenCloseFist**, **ImaginedOpenCloseFist**, **FeetHands**, and **ImaginedFeetHands** (Schalk et al., 2022, 2004) are all datasets collected from 109 participants that involve BCI based motor imagery of hand and feet movement. The task is to predict the real or imagined movement. The first two tasks involved opening and closing either the participant's left or right fist, with the first being real movements (OpenCloseFist) and the second imagined movements (ImaginedOpenCloseFist). The third and fourth involve predicting whether the subject is closing their hand into a fist or curling their toes, either real (FeetHands) or imagined (ImaginedFeetHands). Each subject performed three sessions of two minutes for each task. In each session the subject was shown an image on a screen in various locations, and moved or imagined moving a corresponding body part in reaction. This repeated 15 times in each run, leading to each task being completed 45 times per participant.

The data was recorded at 160Hz with 64 EEG channels using the 10-10 system. Each task was separated into its own classification problem, creating four distinct datasets. For each, a four second window was taken around the completion of each task, starting one second before initial onset. Four subjects were also removed due to inconsistent recording frequencies, but no other pre-processing steps have been taken.

**InnerSpeech**, **PronouncedSpeech**, and **VisualSpeech** (Nieto et al., 2021) involve predicting real or imagined speech of one of four words: up, down, left or right. Ten participants completed three recording sessions, each comprising three tasks. In each trial, participants were shown a prompt (a triangle pointing in a direction) and responded in one of three ways: speaking the direction aloud (PronouncedSpeech), imagining speaking it (InnerSpeech), or visualising moving a circle in the indicated direction (VisualSpeech). Each task was repeated multiple times for each of the four direction words, with short breaks between repetitions.

EEG was recorded at 1024 Hz using 128 channels. Each task defines a separate four-class classification problem. For each trial, a 3.5 s window was extracted around the onset of the prompt, starting 0.5 s before it appeared. As the number of repetitions varied across participants, we retained only the first 50 instances per word, per task, per participant. No additional preprocessing was applied.

**LowCost** (Peterson et al., 2022) comes from a BCI experiment to predict resting state vs imagined dominant hand movement. It was designed to investigate performance of lower cost EEG equipment and its viability for BCI studies compared to the more expensive traditional equipment set. 10 subjects completed four sessions, each consisting of 15 occurrences of each class. The data was recorded at 125Hz with 15 EEG channels, with each instance representing three seconds starting 200ms before stimulus.

**MindReading** was a challenge organised in conjunction with the International Conference on Artificial Neural Networks (ICANN) 2011, sponsored by the PASCAL2 Challenge Programme[6]. Unlike the rest of the datasets in the archive, this was recorded using MEG rather than EEG. The data consists of MEG recordings of a single subject, made during two separate measurement sessions on consecutive days. The task was to infer from one-second time windows the type of visual stimulus shown to the subject (without audio). The five class values are: screen savers; clips from nature documentaries; football (clips from La Liga); Mr. Bean; and Chaplin.

**SitStand** (Triana-Guzman et al., 2022) is a BCI dataset for the task of detecting if the subject is sitting, standing or imagining changing from standing to sitting or sitting to standing. 32 participants completed 60 of each task. Due to some irregularities with certain trials we keep only the first 50 instances of each class for each participant. We segment each trial to a 5 second window starting 200ms before stimulus. The data was recorded at 250Hz with 17 EEG channels.

### 2.3 Psychology Based Classification

In psychology research EEG is used to investigate phenomena such as cognitive function, task engagement, emotion, affective and sleep states. This often involves classification problems, but also includes more general investigations into the theory of the mind and more specific studies into before and after behaviour.

**MatchingPennies** (Appelhoff et al., 2018) task involves predicting which hand a subject raised during an interactive game. Subjects were asked to raise either their left or right hand whilst actively imagining raising the other hand. The data was collected from 7 subjects, each raising their left and right hand 100 times. The data was recorded at 5000Hz, with each instance starting 500ms before initial stimulus and lasting 2 seconds, and contains 10 EEG channels.

**FeedbackButton** is the same task as ButtonPress: detecting if a button was pressed or not. However, in this experiment participants were shown a basic prompt to which they would have to respond by either pressing or not pressing the button. The participants were then presented with either positive or negative feedback based on their response, with a goal of optimising the total positive outcomes. Our goal is to detect the thought process that leads to either outcome. The dataset consists of 57 individuals and was recorded at 500Hz with 61 EEG channels. Each classification instance is 900ms long, or 450 data points, starting 200ms before the initial stimulus was shown to the individual, but ending before the actual press occurs.

**SongFamiliarity** (Girard et al., 2025) looks at how the brain can recognise music. 29 participants listened to pieces of music and were asked to press a button when the music sounded familiar. They were then asked to pick the name of the song from a list of four, and were given visual feedback if the correct option was picked. We select a three second window from just before the button press, keeping the first 50 songs each participant listened to. From this we form a classification problem based on whether the song was correctly picked or not. The data was recorded at 1000Hz with 32 EEG channels.

---

6. http://urn.fi/URN:ISBN:978-952-60-4456-9

**Sleep** is a 1-lead EEG dataset that contains 153 whole-night sleeping EEG recordings from 82 healthy subjects. It was formatted as a classification problem in (Zhang et al., 2022). The series were segmented into non overlapping sub series, each of which forms a case. Each case is labelled with one of the five sleeping patterns/stages: Wake (W), Non-rapid eye movement (N1, N2, N3) and Rapid Eye Movement (REM).

Table 1: Overview of EEG datasets used in experiments. N/A means that the subject information was not available from the data.

| Dataset | Channels | Classes | Length | Subjects | Train Size | Test Size | Frequency (Hz) | Data Type |
|---|---|---|---|---|---|---|---|---|
| Alzheimers | 19 | 3 | 30000 | 88 | 45 | 43 | 500 | Med |
| Blink | 4 | 2 | 510 | N/A | 500 | 450 | 255 | BCI |
| ButtonPress | 32 | 2 | 1000 | 1 | 120 | 120 | 1000 | BCI |
| EpilepticSeizures | 1 | 2 | 178 | N/A | 80 | 11420 | 178 | Med |
| EyesOpenShut | 14 | 2 | 128 | N/A | 56 | 42 | 128 | BCI |
| FaceDetection | 144 | 2 | 62 | N/A | 5890 | 3524 | 250 | Psy |
| FingerMovements | 28 | 2 | 400 | N/A | 316 | 100 | 128 | BCI |
| HandMovementDirection | 10 | 4 | 400 | N/A | 160 | 74 | 400 | BCI |
| ImaginedOpenCloseFist | 64 | 2 | 640 | 105 | 2835 | 1890 | 160 | BCI |
| ImaginedFeetHands | 64 | 2 | 640 | 105 | 2835 | 1935 | 160 | BCI |
| InnerSpeech | 128 | 4 | 3584 | 9 | 250 | 200 | 1024 | BCI |
| FibroLiverpool | 64 | 2 | 15360 | 36 | 15 | 21 | 512 | Med |
| LongIntervalTask | 63 | 2 | 4750 | 139 | 1660 | 1120 | 500 | Med |
| LowCost | 15 | 2 | 375 | 10 | 600 | 600 | 125 | BCI |
| MatchingPennies | 10 | 2 | 10000 | 7 | 700 | 700 | 5000 | Psy |
| MindReading | 204 | 5 | 200 | N/A | 727 | 653 | 200 | BCI |
| MotorImagery | 64 | 2 | 3000 | N/A | 278 | 100 | 1000 | BCI |
| OpenCloseFist | 64 | 2 | 640 | 105 | 2835 | 1890 | 160 | BCI |
| FeetHands | 64 | 2 | 640 | 105 | 2835 | 1890 | 160 | BCI |
| PhotoStimulation | 19 | 3 | 9000 | 73 | 37 | 36 | 500 | Med |
| PronouncedSpeech | 128 | 4 | 3584 | 9 | 250 | 200 | 1024 | BCI |
| FeedbackButton | 61 | 2 | 450 | 57 | 1700 | 1150 | 500 | Psy |
| SelfRegulationSCP1 | 6 | 2 | 896 | N/A | 268 | 293 | 256 | BCI |
| SelfRegulationSCP2 | 7 | 2 | 1152 | N/A | 200 | 180 | 256 | BCI |
| ShortIntervalTask | 63 | 2 | 2750 | 139 | 1660 | 1120 | 500 | Med |
| SitStand | 17 | 4 | 1250 | 32 | 3200 | 3200 | 250 | BCI |
| Sleep | 1 | 5 | 178 | 82 | 2394 | 1205 | 100 | Psy |
| SongFamiliarity | 32 | 2 | 3000 | 29 | 700 | 750 | 1000 | Psy |
| FibroUEA | 64 | 2 | 15000 | 41 | 17 | 23 | 500 | Med |
| VisualSpeech | 128 | 4 | 3584 | 9 | 250 | 200 | 1024 | BCI |

## 3 EEG Classification Algorithms

Whilst classification remains a frequent task in EEG research, most studies develop custom solutions to their own dataset which cannot be applied generally to a wider range of problems, or utilise basic classification methods in combination with more advanced feature extraction methods without consideration for newer alternatives (Alarcão and Fonseca, 2019). This is usually the best approach for a specific problem, but it is labour intensive and often required some level of expert knowledge about the task. Our interest is in assessing general purpose classifiers for EEG tasks that span many research fields. Assessing general purpose algorithms offers the potential for faster development and more thorough benchmarking. Different fields tend to have their own preferred techniques. Signal process-

ing approaches usually involve a complex pipeline of transforms used in conjunction with a general purpose classifier. BCI applications tend to use algorithms based around Riemannian distances. As in all fields, deep learning approaches are popular. We summarise some of these algorithms before evaluating them on our archive of problems.

### 3.1 Standard Classifiers

Common features include spectral power in specific frequency bands, time–frequency methods such as wavelet transforms (Subasi, 2007) or summary measures such as entropy, variance or higher-order moments (Tremmel et al., 2024). In BCI, spatial filtering methods such as Common Spatial Patterns (CSP) (Ramoser et al., 2000) are often applied to multi-channel EEG before feature extraction. EEG classification based on Riemannian distance (Yger et al., 2017) using, for example, the Minimum Distance to Mean (MDM) classifier (Barachant et al., 2013) is popular in BCI.

Support Vector Machines (SVM) are the most popular standard method of classification for EEG data (Sha'abani et al., 2020) either on the raw data or on derived features. Random Forest (Jamunadevi et al., 2022; Edla et al., 2018) and Logistic Regression (Singh et al., 2024; Chen et al., 2020) are also popular. One popular feature creation approach is Common Spatial Patterns (CSP) (Antony et al., 2022). CSP is a common signal processing method to extract features using spatial filters. These filters aim to maximise the variance between the classes in the filtered signal, and are fitted to channel-wise covariance matrices. Their use in EEG comes from their ability to separate spatial information in the data, such as between different regions of the brain (Blankertz et al., 2008). For our experiments we combine CSP with SVM in a pipeline.

### 3.2 Deep Learning Classifiers

Deep learning, particularly convolutional neural networks (CNNs), has become a dominant approach for EEG classification (Craik et al., 2019) and is frequently recommended as a general-purpose modelling choice for EEG tasks (Al-Saegh et al., 2021; Rakhmatulin et al., 2024). CNNs learn hierarchies of features by applying convolutional filters to the input, typically interleaving convolution and pooling operations, and training the model end-to-end via gradient-based optimisation.

Several CNN architectures have been proposed specifically for EEG, often motivated by traditional signal-processing pipelines. Two widely used examples are EEGNet (Lawhern et al., 2018) and DeepConvNet (Schirrmeister et al., 2017), which combine temporal filtering with learnable spatial filtering across channels. These models have been applied across a range of EEG classification settings, including sleep staging (Masad et al., 2024), emotion recognition (Cheng et al., 2024), and cognitive load estimation (Pulver et al., 2023).

More recent EEG architectures increasingly use attention mechanisms to model channel structure explicitly, either by learning channel-wise weights that emphasise informative electrodes (Tao et al., 2023; Miao et al., 2023), or by using graph attention to learn inter-channel connectivity for spatio-temporal modelling in applications such as seizure detection and prediction (He et al., 2022; Wang et al., 2023). These approaches are complementary to CNN-based spatial filtering and further highlight the importance of modelling cross-channel structure in EEG.

Given the prominence of deep learning in EEG classification, we include four deep learning baselines in our experiments. We evaluate (i) a standard **CNN** architecture (Zhao et al., 2017) that has been used as a representative deep baseline in comparative time series studies (Fawaz et al., 2019), (ii) **InceptionTime** (Fawaz et al., 2020), a competitive CNN variant based on inception modules, and two EEG-specific architectures, (iii) **EEGNet** (Lawhern et al., 2018) and (iv) **DeepConvNet** (Schirrmeister et al., 2017). For EEGNet and DeepConvNet we use the implementations provided by the Braindecode Python package (Aristimunha et al., 2025) to prioritise reproducibility and accessibility. We acknowledge that many additional deep architectures have been proposed, including transformer-based variants, but these are often task-specific, sensitive to training protocol and compute budget, or lack maintained, general-purpose reference implementations suitable for a uniform benchmark across diverse datasets.

### 3.3 Riemannian Distance Based Classifiers

Traditionally, EEG classification approaches have treated EEG-derived features, such as covariance matrices, as linear structures within Euclidean space. However, these features often reside in non-linear spaces known as Riemannian manifolds - curved spaces formed by collections of covariance matrices - where Euclidean methods may not capture the underlying structure. Riemannian geometry addresses this by defining distances on the manifold (e.g., affine-invariant or log-Euclidean) to measure similarity between covariance representations and thereby capture complex cross-channel interactions (Tibermacine et al., 2024). Concretely, for each EEG case with $C$ channels, we compute the $c \times c$ sample covariance matrix $\Sigma$. This covariance representation encodes cross-channel co-variation (second-order spatial structure) within the segment; the resulting symmetric positive definite (SPD) matrices are treated as points on the SPD Riemannian manifold.

On this manifold, Riemannian distances can be used in a nearest-neighbour classifier (**R-KNN**) or in a Minimum Distance to Mean classifier (**R-MDM**), where each class is represented by its Riemannian mean covariance and test segments are assigned to the closest class mean under the chosen metric. These approaches have primarily been used in BCI applications (Barachant et al., 2013; Yger et al., 2017; Congedo et al., 2017), but have also shown strong performance in clinical settings (Mayaud et al., 2016) and for between-participant classification (Korczowski et al., 2015). For our experiments, we use the MDM-based model due to its widespread adoption in EEG analysis.

### 3.4 Time Series Classifiers

Research into general purpose algorithms for time series classification has been ongoing for decades (Bagnall et al., 2017) and has been made more popular by the establishment of data archives (Dau et al., 2019; Ruiz et al., 2021) and open source implementations (Middlehurst et al., 2024a). A recent comparative study (Middlehurst et al., 2024b) found that two algorithms were significantly better than the rest on these archives:

**Multi-ROCKET-Hydra (MRHydra)** (Dempster et al., 2023) is the latest version of a series of classifiers based on feature extraction using randomised convolutions. These features are concatenations of those found using HYbrid Dictionary–ROCKET Architecture

(Hydra) (Dempster et al., 2023) and the Multiple pooling operator Random Convolutional Kernel Transform (MultiROCKET) (Tan et al., 2022).

**HIVE-COTE v2.0** (Middlehurst et al., 2021) is the most recent version of the Hierarchical Vote Collective of Transformation Ensembles (HIVE-COTE) family of classifiers. HC2 is an ensemble of four different time series classification modules, each of which is built using a different representation of the data. The modules are ensembled using the Cross-validation Accuracy Weighted Probabilistic Ensemble (CAWPE) (Large et al., 2019). CAWPE employs a tilted probability distribution using exponential weighting of probabilities estimated for each module found through cross-validation on the train data. The weighted probabilities from each module are summed and standardised to produce the HC2 probability prediction. These modules are the latest versions of the Shapelet Transform Classifier (STC) (Hills et al., 2014), the Temporal Dictionary Ensemble (TDE) (Middlehurst et al., 2020), the Diverse Representation Canonical Interval Forest (DrCIF) and an ensemble of ROCKET classifiers called the Arsenal. For a more detailed description, see (Middlehurst et al., 2021). The classifiers we use and their implementation are presented in Table 2.

Table 2: EEG Classifiers and Corresponding Libraries

| Classifier | Abbreviation | Library/Framework |
|---|---|---|
| Riemannian-MDM | R-MDM | pyriemann |
| Common Spatial Patterns SVM | CSP-SVM | MNE[7] + scikit-learn |
| Convolutional Neural Network | CNN | aeon[8] |
| InceptionTime (Fawaz et al., 2020) | IT | aeon |
| EEGNet (Lawhern et al., 2018) | EEGNet | Braindecode[9] |
| DeepConvNet (Schirrmeister et al., 2017) | DCN | Braindecode[10] |
| HIVE-COTE v2.0 (Middlehurst et al., 2021) | HC2 | aeon |
| Multi-ROCKET Hydra (Dempster et al., 2023) | MRHydra | aeon |

## 4 Results

We train each of the eight classifiers listed in Table 2 on the default 30 train set EEG classification datasets described in Table 1 and evaluate them on the test data.

We assess performance using classification accuracy to measure general performance and balanced accuracy to assess algorithms in the presence of imbalance. The quality of the probability estimates is measured with the log loss. The ability to rank predictions is estimated by the area under the receiver operating characteristic curve (AUROC).

All experiments were carried out with a single processor per experiment on a shared HPC facility. The four deep learning classifiers were run on GPU, the rest on CPU. We compare multiple classifiers over multiple datasets using an adaptation of the critical difference diagram (Demšar, 2006). The post-hoc Nemenyi test is replaced with a comparison of all classifiers using pairwise Wilcoxon signed-rank tests, and cliques formed using the Holm correction as recommended by (Benavoli et al., 2016).

Figure 1 shows the average ranks of the eight classifiers used in the comparison. HC2 is the best performing algorithm and it has significantly better accuracy than all the others.

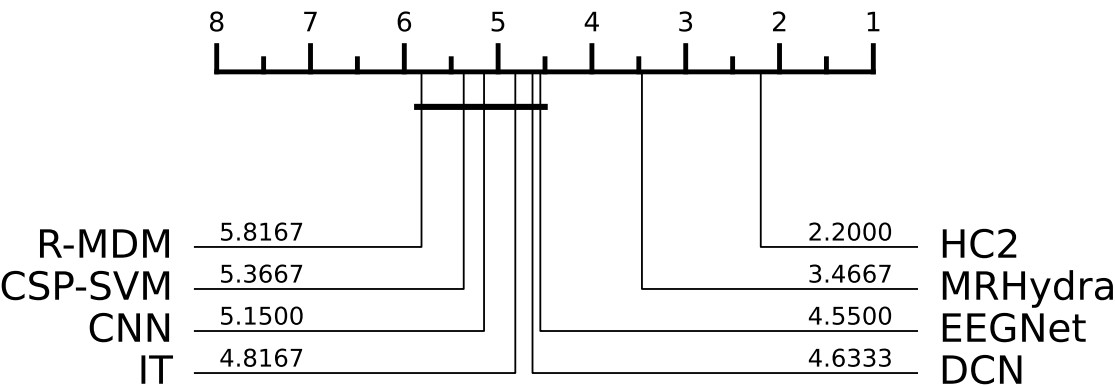

Figure 1: A comparison of the accuracy ranks for eight classifiers on 30 EEG classification problems

MRHydra is the second best, and is significantly better than the other 6 classifiers tested. The remaining 6 classifiers all form the third significance group, with the two non-deep learning methods, R-MDM and CSP-SVM performing the worst, but were not significantly worse than the deep learning group.

| Mean-Accuracy | HC2 0.6049 | MRHydra 0.5889 | IT 0.5421 | CNN 0.5282 | DCN 0.5026 | EEGNet 0.4938 | |
|---|---|---|---|---|---|---|---|
| HC2 0.6049 | Mean-Difference r>c / r=c / r<c Wilcoxon p-value | **0.0159** **21 / 1 / 8** **0.0125** | **0.0628** **23 / 3 / 4** **≤ 1e-04** | **0.0766** **25 / 1 / 4** **0.0001** | **0.1023** **23 / 0 / 7** **0.0006** | **0.1111** **24 / 2 / 4** **0.0002** | 0.10 |
| MRHydra 0.5889 | -0.0159 8 / 1 / 21 0.9875 | - | **0.0468** **21 / 0 / 9** **0.0110** | **0.0607** **21 / 0 / 9** **0.0016** | **0.0864** **18 / 1 / 11** **0.0368** | **0.0951** **17 / 1 / 12** **0.0132** | 0.05 |
| IT 0.5421 | -0.0628 4 / 3 / 23 1.0000 | -0.0468 9 / 0 / 21 0.9896 | - | 0.0139 15 / 2 / 13 0.1827 | 0.0395 13 / 1 / 16 0.3710 | 0.0483 13 / 1 / 16 0.4795 | 0.00 |
| CNN 0.5282 | -0.0766 4 / 1 / 25 0.9999 | -0.0607 9 / 0 / 21 0.9985 | -0.0139 13 / 2 / 15 0.8173 | - | 0.0257 10 / 5 / 15 0.5493 | 0.0344 14 / 2 / 14 0.2963 | -0.05 |
| DCN 0.5026 | -0.1023 7 / 0 / 23 0.9994 | -0.0864 11 / 1 / 18 0.9632 | -0.0395 16 / 1 / 13 0.6290 | -0.0257 15 / 5 / 10 0.4507 | - | 0.0088 10 / 2 / 18 0.5975 | -0.10 |
| EEGNet 0.4938 | -0.1111 4 / 2 / 24 0.9998 | -0.0951 12 / 1 / 17 0.9868 | -0.0483 16 / 1 / 13 0.5205 | -0.0344 14 / 2 / 14 0.7037 | -0.0088 18 / 2 / 10 0.4025 | **If in bold, then p-value < 0.05** | |

Figure 2: A pairwise comparison of the top five classifiers.

The pairwise relative performance of the top five algorithms is shown in Figure 2 and Table 3 shows the average of each algorithm over a range of performance measures. On average, HC2 is over 1.5% more accurate than MRHydra and beats it on 21 out of 30 datasets. It is the best performer on all metrics except for fit time. HC2, CSP-SVM and EEGNet give the best probability estimates (assessed by log loss) and HC2 and IT are best at ranking cases (AUROC). MRHydra gives only 0/1 predictions, hence its large log loss values.

Table 3: Averaged performance metrics across classifiers, timings are run time averaged over the archive in hours, minutes or seconds.

| Metric | HC2 | MRHydra | CNN | IT | EEGNet | DCN | CSP-SVM | R-MDM |
|---|---|---|---|---|---|---|---|---|
| Accuracy | 0.60 | 0.57 | 0.52 | 0.53 | 0.50 | 0.51 | 0.53 | 0.48 |
| Balanced Accuracy | 0.58 | 0.56 | 0.50 | 0.53 | 0.51 | 0.52 | 0.49 | 0.49 |
| AUROC | 0.68 | 0.62 | 0.58 | 0.64 | 0.62 | 0.63 | 0.58 | 0.56 |
| LogLoss | 0.77 | 15.46 | 2.97 | 2.21 | 0.86 | 2.12 | 0.89 | 3.17 |
| Fit Time (hr/min/s) | 78.62 hr | 2.88 min | 9.88 min | 2.98 hr | 2.04 min | 1.88 min | 22.89 s | 0.31 s |
| Predict Time (hr/min/s) | 19.22 hr | 2.29 min | 0.79 s | 5.29 s | 4.29 s | 3.36 s | 4.29 s | 0.99 s |

Table 7 in the appendix shows the test set accuracy for all the classifiers and datasets. The first column presents the accuracy of a baseline classifier that always predicts the majority class. For many of these tasks, the classifiers perform no better than this naive approach. This suggests that, if any discriminative information exists in the signals, it is not readily accessible to standard classifiers and would likely require expert-driven, hand-crafted feature engineering to uncover.

HC2 is the best performer on 17 datasets, MRHydra on 5. Together, they win on 22 out of 30 problems. This demonstrates the potential for TSML approaches as a baseline starting point for EEG classification.

The two EEG-specific deep learning models perform reasonably overall, with EEGNet ranking best on three datasets and DCN on two. Their weaker results on several datasets are plausibly attributable to limited training data: in EEG studies, recruiting participants and collecting high-quality recordings is costly, so many benchmarks remain small by deep learning standards. In such settings, performance can often be improved by strategies that reduce sample complexity, including strong regularisation (dropout, weight decay, early stopping), data augmentation tailored to EEG (time shifts, additive noise, channel dropout, frequency-band perturbations, mixup), transfer learning or pretraining on larger EEG corpora followed by fine-tuning, and evaluation protocols that maximise training usage (e.g., cross-validation and subject-wise splits where appropriate). These strategies require problem specific information. Limited sample size does not fully explain the observed results: on the two fibromyalgia datasets, the deep models perform strongly despite relatively few cases. One plausible explanation is that these recordings are longer, resting-state segments where spectral structure is more stable and informative, aligning well with architectures such as EEGNet that explicitly target frequency-sensitive patterns, whereas many BCI datasets consist of short, cue-locked trials with higher inter-trial and inter-subject variability.

The SVM based classifier is consistent but is only top ranked on one dataset when combined with CSP. This indicates that it is a useful benchmark, but is unlikely to be the best approach. We have not tuned any of the classifiers, and this may improve performance, particularly for SVM based classifiers (Bagnall and Cawley, 2017). However, tuning on small train set sizes can be problematic and it is difficult to make algorithm comparison fair when tuning. HC2 has been shown to be robust to parameter settings (Middlehurst et al., 2021).

The performance of R-MDM is perhaps surprising, given its popularity in the BCI community. We have done all we can to validate these results and have used pyriemann, a well established open source toolkit. This approach is based on covariance matrices. This

means it's focussed on the interaction between channels rather than the observed values. Whilst this will identify regions behaving similarly to stimulus, it discards information about changes in values for any specific channel.

There is no one best approach, and there is significant variation between algorithms on certain data. The average spread of maximum to minimum accuracy is approximately 20%. This demonstrates that the nature of the best performing algorithm may give insights into the problem, based on the nature of the features used. For example, R-MDM is the only algorithm to outperform the baseline for EyesOpenShut. This suggests that correlation between channels is likely the most important feature for this problem. MRHydra is one of two approaches better than default for FibroLiverpool and the best by far on Alzheimers. Convolutions used by MRHydra could potentially be biomarkers for these conditions. These empirical insights might help a domain expert design future experiments.

Overall, HC2 is clearly the best performing algorithm, but this comes at a considerable cost. HC2 is significantly slower than all the other algorithms. We cannot directly compare run times due to different hardware, but on average the picture is very clear. HC2 takes a median of approximately 14 hours on a single CPU, and a mean of 78 hours. The next slowest was InceptionTime, which took 3 hours on average (on a GPU). MRHydra, the second best performing algorithm, is much faster, taking seconds or minutes rather than hours. Full runtimes for all problems and algorithms are presented in Table 8 in the appendix. We investigate the causes of the slow runtime of HC2 in Section 5.1.

As a benchmark, we recommend using HC2 if accuracy is key and computation time is not a concern, and MRHydra as a faster alternative.

## 5 HIVE-COTE Deconstruction

HC2 is a meta-ensemble of four different classifiers, each built on a different representation. The relative performance of the four components and HC2 is shown in Figure 3. One component, TDE, is significantly worse than the others. TDE is a dictionary-based classifier, which means it classifies based on the frequency of short repeating patterns. Among Arsenal, STC, and DrCIF we observe no significant difference in average rank, and HC2 is significantly better than Arsenal and STC but not significantly better than DrCIF. HC2 performs well because it combines the diversity of different approaches in a structured way. There is a high degree of diversity between the components. Figure 4 illustrates the performance diversity between ensemble components. Figure 4(a) plots the shapelet-based STC against the convolution-based Arsenal. Whilst there is no overall difference in performance, the range of accuracies is large: the mean absolute difference in accuracy across datasets exceeds 5 percentage points.

The strength of HC2 is that it is robust, i.e. it achieves good performance when the best discriminatory features are embedded in any one of the component representations. However, it only improves on the best performing component on seven of the 30 datasets. This demonstrates the potential of HC2 as an exploratory tool: if one component is more accurate than others, then investigation into the internal representation may lead to insights into the problem.

HC2 is the best algorithm on average; however, across the archive we observe variation that underlines why no single representation is universally optimal. To explore relative

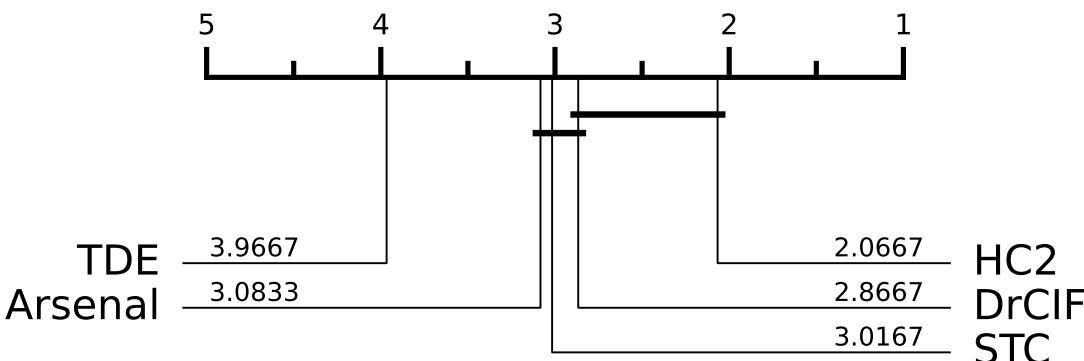

Figure 3: A comparison of the accuracy ranks for HC2 and its four components.

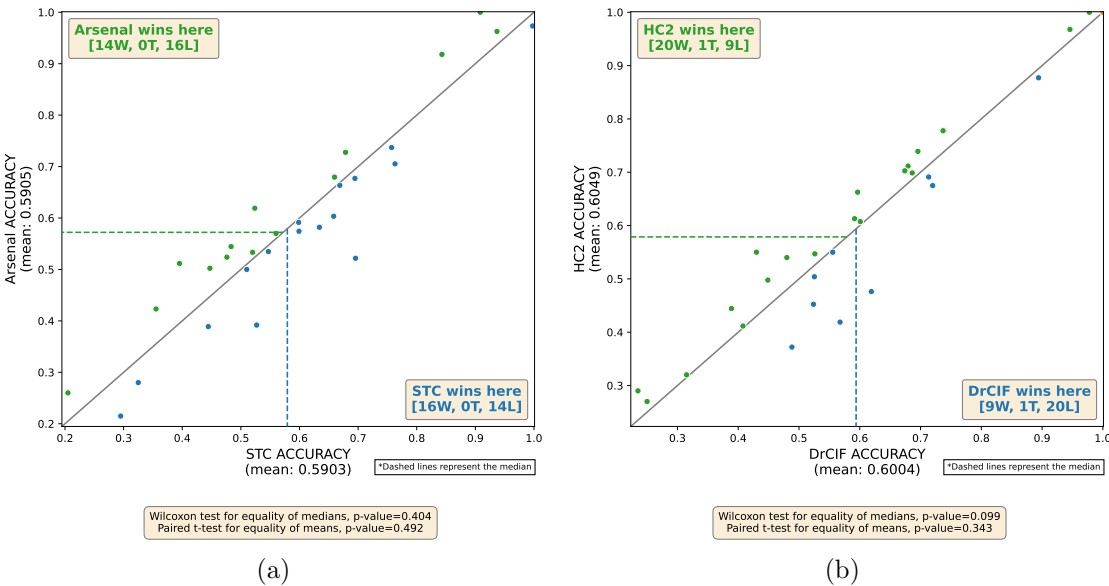

Figure 4: Scatter plot of accuracies for the classifiers STC vs Arsenal (a) and HC2 vs DrCIF (b).

performance more closely, we define failure-gap datasets as those where the best TSML method is at least 10 percentage points less accurate than the best non-TSML baseline, summarised in Table 4.

For **HandMovementDirection**, HC2 performs poorly relative to CSP-SVM (41.89% vs 60.81%), suggesting that explicitly modelling spatial structure through supervised spatial filtering is important for this task and that generic TSML representations can miss discriminative cross-channel patterns. This is also reflected in the error structure: Table 5 shows

Table 4: Largest failure-gap case studies, defined as datasets where the best TSML method (HC2) is at least 10 percentage points less accurate than the best non-TSML baseline.

| Dataset | Best TSML | Acc. (%) | Best non-TSML | Acc. (%) | Gap (pp) |
|---|---|---|---|---|---|
| HandMovementDirection | HC2 | 41.89 | CSP-SVM | 60.81 | 18.92 |
| FibroUEA | HC2 | 73.91 | DeepConvNet | 86.96 | 13.05 |
| EyesOpenShut | HC2 | 45.24 | R-MDM | 57.14 | 11.90 |

Table 5: Confusion matrices for HandMovementDirection comparing HC2 and CSP-SVM.

| | **HC2** | | | | | **CSP-SVM** | | | |
|---|---|---|---|---|---|---|---|---|---|
| True\Pred | 0 | 1 | 2 | 3 | True\Pred | 0 | 1 | 2 | 3 |
| 0 | 3 | 3 | 6 | 3 | 0 | 8 | 0 | 2 | 5 |
| 1 | 1 | 16 | 7 | 6 | 1 | 2 | 14 | 12 | 2 |
| 2 | 0 | 4 | 9 | 2 | 2 | 0 | 2 | 11 | 2 |
| 3 | 0 | 4 | 7 | 3 | 3 | 0 | 2 | 0 | 12 |

that HC2 produces substantial confusion across classes, particularly for class 0 (only 3/15 correct) and frequent mislabelling into classes 2 and 3, whereas CSP-SVM reduces these confusions and improves recognition of class 3 (12 correct) while substantially increasing correct predictions for class 0 (8 correct).

For **EyesOpenShut**, the covariance-based Riemannian MDM classifier leads (57.14% vs 45.24%), indicating that second-order cross-channel statistics are discriminatory. Table 6 (left) illustrates that both approaches achieve the same performance on class 0, while the improvement of R-MDM is driven by better detection of class 1 (6 correct vs 1 for HC2). Finally, for **FibroUEA**, the EEG-specific deep model DeepConvNet substantially outperforms HC2 (86.96% vs 73.91%). Table 6 (right) shows that HC2 defaults to the majority class in all but one case, whereas DeepConvNet produces a more balanced set of predictions and improves detection of the minority class.

Table 6: Confusion-matrix case studies comparing HC2 against task-appropriate baselines. Left: EyesOpenShut (HC2 vs R-MDM). Right: FibroUEA (HC2 vs DeepConvNet).

| | **EyesOpenShut** | | | | | **FibroUEA** | | | |
|---|---|---|---|---|---|---|---|---|---|
| | **HC2** | | **R-MDM** | | | **HC2** | | **DeepConvNet** | |
| True\Pred | 0 | 1 | True\Pred | 0 | 1 | True\Pred | 0 | 1 | True\Pred | 0 | 1 |
| 0 | 18 | 3 | 0 | 18 | 3 | 0 | 1 | 6 | 0 | 5 | 2 |
| 1 | 20 | 1 | 1 | 15 | 6 | 1 | 0 | 16 | 1 | 1 | 15 |

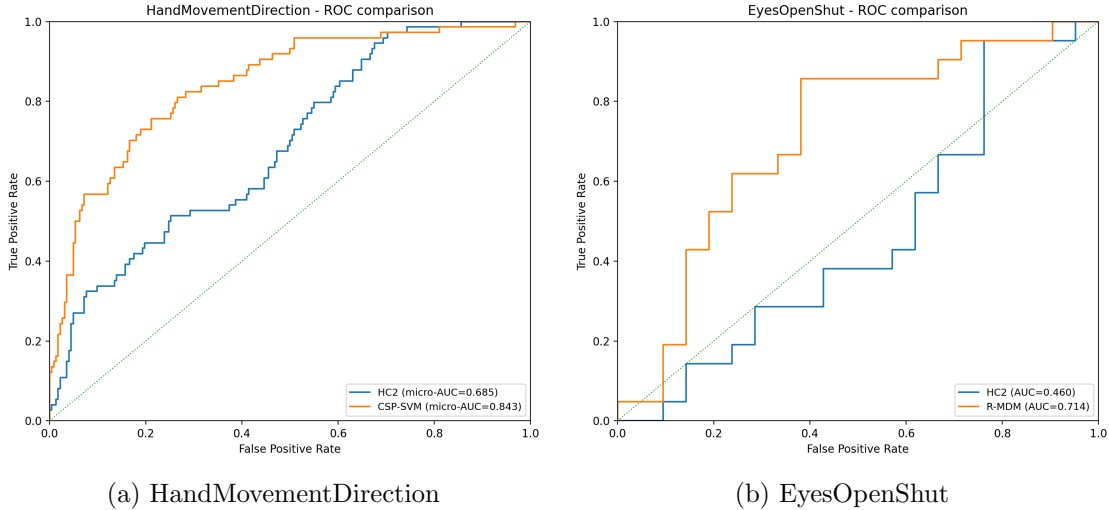

(a) HandMovementDirection

(b) EyesOpenShut

Figure 5: ROC comparisons for two case-study datasets. (a) HandMovementDirection, comparing HC2 and CSP-SVM (micro-AUC 0.685 vs 0.843). (b) EyesOpenShut, comparing HC2 and R-MDM (AUC 0.460 vs 0.714).

Figure 5 complements the accuracy and confusion-matrix case studies by comparing the quality of the probabilistic ranking produced by each method across thresholds. For **HandMovementDirection**, CSP-SVM achieves a substantially higher micro-AUC than HC2 (0.843 vs 0.685), indicating stronger separation between the true class and competing classes and aligning with the reduced cross-class confusion seen for CSP-SVM. For **EyesOpenShut**, R-MDM attains an AUC of 0.714, whereas HC2 is below 0.5 (0.460), suggesting that covariance-based representations capture much of the discriminative signal for this dataset.

## 5.1 HC2 Runtime

The main practical limitation of HC2 is its computational cost, which is driven by its constituent classifiers. We ran HC2 single-threaded on a CPU to maintain a controlled and comparable hardware setting. In practice, HC2 can be accelerated substantially because its constituent classifiers are independent and can be trained in parallel. Several components also support internal parallelism (e.g., across trees or base estimators) in the aeon reference implementations.

With default settings, STC and DrCIF are the slowest components to train. For STC, the dominant cost is the rotation forest classifier applied after the shapelet transform; it involves repeated matrix inversions as part of a piecewise decomposition. In principle, reducing the post-transform feature set (or the size of the rotation forest) could lower computation with limited impact on accuracy. DrCIF is an interval-based tree ensemble that fits 200 trees by default, and reducing the number of trees would directly reduce training time.

A further opportunity is model pruning. On these EEG datasets, TDE is consistently the slowest component at inference because it is nearest-neighbour based. It also contributes

relatively little to the ensemble's accuracy in our results, suggesting that omitting TDE (or replacing it with a cheaper spectral- or covariance-based component) could yield a meaningful reduction in inference time with limited loss of accuracy. HC2 also supports *contracting*, in which a time budget is specified and each constituent is built under an approximate runtime constraint (for example, by reducing the number of estimators or limiting candidate features). We do not explore these acceleration options here, as our goal is to provide a first, conservative benchmark of the algorithms in their standard forms. The runtimes reported in this paper should therefore be interpreted as upper bounds on HC2 training time when parallelism and contracting are enabled.

At present, HC2 and its constituent classifiers do not have GPU implementations in the reference aeon toolkit. Where applicable, GPU acceleration would be expected to reduce training time by parallelising the most expensive feature-extraction and model-fitting operations.

## 6 Conclusions

In this paper, the primary goal was to introduce the first comprehensive repository of EEG classification problems and evaluate a range of general-purpose time series classifiers alongside established deep learning and domain-specific approaches. We collate and form an archive of 30 EEG datasets from a range of different fields, containing datasets with a variety of characteristics including number of participants, channel counts, classes, and frequency. This range means that classifiers tested on this archive can be evaluated on their ability to generalise across EEG problems, something that we believe to be missing in current EEG research.

The second goal was to provide benchmark results on this archive, which future studies can compare against. Our experiments show that HIVE-COTE v2.0 (HC2) (Middlehurst et al., 2021), a state-of-the-art TSC algorithm, substantially outperforms both standard machine learning methods and deep learning models.

HC2 achieves the highest accuracy overall, but at a significant computational cost. MRHydra, while slightly less accurate, offers a more computationally efficient alternative. The relatively inferior performance of the deep learning models can likely be attributed to the limited size of EEG datasets, a common constraint in domains such as clinical diagnosis. This highlights a key limitation of data-hungry methods in EEG research and motivates the development of models better suited to low-resource settings. HC2 is relatively slow when using the default configuration for EEG. This can be improved, but we would also note that collecting EEG data is time consuming and can be expensive if it involves formal trials: if it takes a day to train the classifier, this is unlikely to delay the project overall, although runtimes such as seen on ShortIntervalTask (15 days) are likely to present a bottleneck.

The results validate HC2 and MRHydra as robust baselines for EEG classification tasks. While we did not customise these classifiers, our analysis highlights avenues for improving performance and efficiency. For example, adapting HC2 through channel selection or replacing individual components may reduce training time. Furthermore, HC2 primarily looks at information contained within the temporal domain. Including features extracted from the spatial and spectral domains may be key for optimising classification accuracy. Spatial information could be included by incorporating inter-channel interactions, a feature largely

absent in current TSML models. This may enhance performance on tasks that depend on spatial relationships between EEG electrodes. Spectral features are harder to extract from BCI problems due to short, segmented time series, but could be vital for longer resting-state data.

We recommend the use of HC2, MRHydra, CSP-SVM, and possibly EEGNet as baseline comparators in future EEG classification studies. HC2 is preferred when performance is paramount and the computational cost is acceptable, while MRHydra provides a good balance between speed and accuracy. CSP-SVM can be used to give a fast baseline measure with good probability estimates. EEGNet can give an indication of whether spectral features might be key for that specific problem, but can perform significantly worse on other problems so should not be used alone.

However, we recognise that our evaluation does not include all methods for EEG classification, focusing on the most commonly used in general research. Therefore, future work would involve comparing an optimised version of HC2 to some of these newer, more specialised EEG classification methods, such as Medformer (Wang et al., 2024) and GREEN (Paillard et al., 2025).

## Acknowledgments

This work has been supported by the UK Research and Innovation Engineering and Physical Sciences Research Council (grant reference EP/W030756/2). The authors acknowledge the use of the High Performance Computing Cluster supported by the Research and Specialist Computing Support service at the University of East Anglia and the IRIDIS High Performance Computing Facility, and associated support services at the University of Southampton. We would like to thank all those contributing to open-source EEG datasets in the archive and the implementations of the algorithms used in the experiments.

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

## 7 Appendix

Table 7: EEG Classification Accuracy with Best Results in Bold. Default is the accuracy for predicting the majority class.

| Estimators: | Default | HC2 | MRHydra | CNN | IT | EEGNet | DCN | CSP-SVM | R-MDM |
|---|---|---|---|---|---|---|---|---|---|
| Alzheimers | 42% | 37.21% | **51.16%** | 41.86% | 37.21% | 32.56% | 41.86% | 39.53% | 39.53% |
| Blink | 56% | **100.00%** | 97.78% | 78.44% | 73.11% | 83.11% | 52.22% | 87.33% | 32.89% |
| ButtonPress | 56% | **100.00%** | **100.00%** | 70.83% | 98.33% | 44.17% | 55.00% | 67.50% | 95.00% |
| EpilepticSeizures | 80% | **96.79%** | 96.27% | 86.79% | 95.97% | 17.73% | 34.33% | 95.28% | 86.65% |
| EyesOpenShut | 50% | 45.24% | 42.86% | 50.00% | 45.24% | 50.00% | 50.00% | 33.33% | **57.14%** |
| FaceDetection | 50% | 66.26% | 59.14% | 65.72% | 67.17% | 59.25% | **67.62%** | 53.66% | 51.53% |
| FeedbackButton | 50% | **54.70%** | 55.65% | 49.83% | 49.30% | 52.78% | 52.70% | 46.96% | 50.78% |
| FeetHands | 50% | **70.26%** | 64.71% | 50.42% | 63.86% | 68.68% | 66.93% | 55.50% | 49.68% |
| FibroLiverpool | 52% | 47.62% | **71.43%** | 47.62% | 52.38% | **71.43%** | 52.38% | 47.62% | 47.62% |
| FibroUEA | 70% | 73.91% | 65.22% | 69.57% | 39.13% | 52.17% | **86.96%** | 69.57% | 47.83% |
| FingerMovements | 51% | 54.00% | 58.00% | 49.00% | **56.00%** | 54.00% | 52.00% | 51.00% | 50.00% |
| HandMovementDirection | 41% | 41.89% | 32.43% | 59.46% | 41.89% | 33.78% | 32.43% | **60.81%** | 40.54% |
| ImaginedFeetHands | 50% | **69.87%** | 63.57% | 67.08% | 63.31% | 67.80% | 67.65% | 54.52% | 49.87% |
| ImaginedOpenCloseFist | 50% | **77.78%** | 70.21% | 50.05% | 73.60% | 77.25% | 76.93% | 58.15% | 50.69% |
| InnerSpeech | 26% | **27.00%** | 25.50% | 23.50% | 23.00% | 22.50% | 25.00% | 26.00% | 26.00% |
| LongIntervalTask | 68% | **71.16%** | 69.29% | 64.11% | 69.46% | 32.68% | 32.14% | 67.86% | 60.09% |
| LowCost | 50% | **67.50%** | 58.50% | 65.00% | 64.00% | 54.67% | 61.33% | 50.33% | 50.33% |
| MatchingPennies | 50% | **61.29%** | 59.43% | 50.29% | 52.00% | 60.86% | 56.43% | 58.86% | 57.00% |
| MindReading | 23% | **49.77%** | 41.19% | 23.12% | 19.14% | 22.21% | 19.91% | 35.83% | 41.04% |
| MotorImagery | 50% | **55.00%** | 51.00% | 50.00% | 54.00% | **55.00%** | 50.00% | 47.00% | 53.00% |
| OpenCloseFist | 50% | 77.78% | 70.79% | 50.05% | 75.19% | **82.17%** | 80.42% | 61.06% | 51.32% |
| PhotoStimulation | 42% | **44.44%** | 38.89% | 41.67% | 22.22% | 33.33% | 41.67% | 38.89% | 33.33% |
| PronouncedSpeech | 26% | **32.00%** | 27.00% | 24.00% | 23.50% | 28.00% | 26.50% | 25.50% | 25.00% |
| SelfRegulationSCP1 | 50% | 87.71% | **94.88%** | 84.30% | 84.30% | 84.30% | 82.59% | 75.77% | 57.34% |
| SelfRegulationSCP2 | 50% | **55.00%** | 53.33% | 48.33% | 50.00% | 47.78% | 49.44% | 50.00% | 49.44% |
| ShortIntervalTask | 64% | 69.11% | **74.29%** | 64.55% | 69.20% | 35.89% | 35.71% | 64.29% | 58.75% |
| SitStand | 25% | 41.16% | **41.97%** | 32.34% | 33.69% | 28.09% | 27.84% | 26.13% | 34.03% |
| Sleep | 49% | **60.75%** | 57.43% | 48.88% | 55.77% | 50.29% | 50.29% | 54.52% | 33.68% |
| SongFamiliarity | 52% | 50.40% | 51.33% | 51.87% | 48.27% | **52.40%** | 51.87% | 50.27% | 51.47% |
| VisualSpeech | 26% | **29.00%** | 23.50% | 26.00% | 26.00% | 26.50% | 27.50% | 26.00% | 24.00% |
| Wins | | 17.0 | 5.0 | 0.0 | 1.0 | 3.0 | 2.00 | 1.00 | 1.00 |

Table 8: Average fit time (units shown as s, min, or hr)

| Estimators: | HC2 | MRHydra | CNN | IT | EEGNet | DCN | CSP-SVM | R-MDM |
|---|---|---|---|---|---|---|---|---|
| Alzheimers | 16.41 hr | 12.27 s | 53.81 s | 1.57 hr | 1.99 s | 3.24 s | 0.25 s | 0.01 s |
| Blink | 1.27 hr | 18.28 s | 2.53 min | 39.63 min | 1.84 s | 6.79 s | 0.53 s | 0.03 s |
| ButtonPress | 1.51 hr | 1.95 s | 3.38 min | 19.06 min | 0.65 s | 0.65 s | 0.07 s | 0.03 s |
| EpilepticSeizures | 2.78 min | 2.08 s | 37.85 s | 6.61 min | 0.13 s | 0.14 s | 0.01 s | 0.72 s |
| EyesOpenShut | 5.35 min | 1.61 s | 22.29 s | 41.13 min | 0.10 s | 0.16 s | 0.01 s | 0.01 s |
| FaceDetection | 488.42 hr | 1.77 min | 30.33 min | 2.96 hr | 1.73 min | 1.23 min | 18.36 s | 8.95 s |
| FeedbackButton | 29.26 hr | 1.18 min | 17.23 min | 1.83 hr | 1.06 min | 1.07 min | 13.01 s | 0.70 s |
| FeetHands | 127.78 hr | 3.53 min | 28.82 min | 4.31 hr | 2.68 min | 1.91 min | 55.41 s | 1.18 s |
| FibroLiverpool | 9.19 hr | 43.98 s | 36.02 s | 55.51 min | 1.49 s | 1.27 s | 0.03 s | 0.02 s |
| FibroUEA | 10.50 hr | 52.69 s | N/A | N/A | 2.80 s | 1.62 s | 0.03 s | 0.02 s |
| FingerMovements | 45.34 min | 3.45 s | 1.68 min | 12.43 min | 0.77 s | 1.37 s | 0.05 s | 0.02 s |
| HandMovementDirection | 26.41 min | 6.72 s | 1.41 min | 12.53 min | 1.04 s | 3.20 s | 0.08 s | 0.01 s |
| ImaginedFeetHands | 103.33 hr | 3.56 min | 17.82 min | 4.27 hr | 2.68 min | 1.92 min | 52.05 s | 1.22 s |
| ImaginedOpenCloseFist | 97.75 hr | 3.50 min | 17.58 min | 4.26 hr | 2.66 min | 2.78 min | 55.23 s | 1.18 s |
| InnerSpeech | 21.53 hr | 2.72 min | 4.75 min | 1.92 hr | 1.95 min | 1.25 min | 1.55 s | 0.83 s |
| LongIntervalTask | 326.79 hr | 19.92 min | 22.62 min | 15.50 hr | 17.31 min | 14.41 min | 1.45 min | 0.61 s |
| LowCost | 2.93 hr | 19.31 s | 6.73 min | 36.69 min | 3.90 s | 12.59 s | 0.54 s | 0.07 s |
| MatchingPennies | 116.16 hr | 12.31 min | 7.63 min | 13.06 hr | 1.77 min | 4.69 min | 34.79 s | 0.06 s |
| MindReading | 1.13 hr | 1.70 s | 7.48 min | 30.29 min | 52.68 s | 50.93 s | 0.47 s | 8.44 s |
| MotorImagery | 14.63 hr | 2.00 min | 4.17 min | 1.72 hr | 1.13 min | 55.08 s | 1.85 s | 0.07 s |
| OpenCloseFist | 97.49 hr | 3.71 min | 28.27 min | 4.32 hr | 2.66 min | 2.78 min | 49.53 s | 1.17 s |
| PhotoStimulation | 7.18 hr | 39.47 s | 50.70 s | 1.09 hr | 1.12 s | 1.24 s | 0.09 s | 0.01 s |
| PronouncedSpeech | 21.36 hr | 2.47 min | 5.26 min | 1.93 hr | 1.93 min | 2.31 min | 1.65 s | 0.81 s |
| SelfRegulationSCP1 | 1.30 hr | 20.11 s | 3.19 min | 33.24 min | 1.68 s | 5.05 s | 0.27 s | 0.02 s |
| SelfRegulationSCP2 | 1.48 hr | 18.43 s | 2.71 min | 32.00 min | 3.24 s | 4.63 s | 0.22 s | 0.02 s |
| ShortIntervalTask | 352.34 hr | 9.68 min | 20.38 min | 9.29 hr | 9.72 min | 7.95 min | 53.76 s | 0.62 s |
| SitStand | 210.25 hr | 6.08 min | 30.63 min | 8.23 hr | 1.66 min | 2.48 min | 2.95 min | 0.66 s |
| Sleep | 5.14 hr | 5.06 s | 25.10 min | 1.21 hr | 1.78 s | 2.91 s | 2.86 s | 0.19 s |
| SongFamiliarity | 49.08 hr | 4.11 min | N/A | N/A | 2.14 min | 2.32 min | 12.84 s | 0.15 s |
| VisualSpeech | 21.61 hr | 2.41 min | N/A | N/A | 2.69 min | 1.23 min | 1.63 s | 0.82 s |

