# OpenReview forum: "Time Series Machine Learning for Classifying Electroencephalograms"
_DMLR — Accepted by DMLR_

### Review · Reviewer_353z · 2025-10-28

**Recommendation:** 3
**Confidence:** 2

**Summary Of Contributions:**

The authors offer a thorough empirical comparison of general-purpose time series machine learning (TSML) algorithms for electroencephalogram (EEG) classification. This work makes two significant contributions:

1. The authors developed and provided a new, large-scale EEG classification archive consisting of 30 datasets from diverse application domains, including clinical assessment, brain-computer interfaces (BCI), and psychology.

2. The authors present a standardized evaluation of a range of classification algorithms, including traditional EEG algorithms, deep learning methods, classifiers based on Riemannian geometry, and a variety of TSML research to date. The authors identify a meta-ensemble algorithm that performs best across the entire benchmark, HIVE-COTE v2.0 (HC2).

**Strengths:**

Significance of contribution: The study provides a strong empirical justification for assessing EEG classification with TSML. It bridges a critical gap between EEG-specific feature pipelines and domain-invariant time series models, allowing reproducible benchmarking.

Relation to prior work: Continues the development of EEG-BCI datasets (e.g., BCI Competition, PhysioNet, Kaggle) and TSC literature (Middlehurst et al., 2021; Ruiz et al., 2021) while enhancing its development with sub-domain diversity and standardization in metadata.

Relevance to the research community: Highly relevant to the intersection of biomedical signal processing, machine learning, and neurotechnology. The archive and codebase will likely be a useful resource for the community.

Quality of the research: The methods are reasonable. The experimental design was appropriate, the metrics were suitable, and the statistical analysis was robust.

Clarity: The writing style is clear, technical, and precise. The figures and the tables meaningfully convey comparison results.

Ethical and social implications: The ethics are fair. The data is from publicly available, encrypted data repositories.

**Audience:**

Yes

**Broader Impact Concerns:**

The paper does not raise any major ethical concerns. The datasets used are open-source and appear to have been collected with appropriate ethical considerations. The development of more accurate EEG classification algorithms has the potential for a significant positive impact in medical diagnostics and assistive technologies.

**Claims And Evidence:**

The claims made in the submission are well-supported by the evidence presented. The authors provide a detailed description of their experimental setup, and the results are presented clearly. The conclusion that HIVE-COTE v2.0 is the best-performing classifier on their benchmark is backed by the experimental results.

**Datasets And Benchmarks:**

The authors have made a valuable contribution by introducing a new EEG classification archive. The authors clearly describe the datasets and have made them available on GitHub, which is advantageous for reproducibility. The authors also adopt the valuable practice of providing default train/test splits, which will lead to fair comparisons of future algorithms.

**Extended Submissions:**

This submission does not appear to be an extended version of a previously published work.

**Limitations:**

Limitations:

- The training time associated with HC2 renders it problematic for large-scale or real-time EEG content analysis.

- EEG repositories differ extensively in terms of the number of channels, frequency, and preprocessing, which is difficult to standardize across such variability.

- TSML methods can perform accurately but remain black boxes, and the relationship to neurophysiological mechanisms has not been evaluated.

- The absence of modern transformer or attention-based models may under-represent the competitiveness of efficient deep learning architectures.

**Requested Changes:**

Critical:

- Expand on the computational complexity of the algorithms studied, especially HIVE-COTE v2.0. This could involve a table comparing training time and inference time.

To strengthen the work:

- Discuss the effects of different pre-processing methods on classifier performance.

- Include a detailed description of the datasets in the archive (for example, describe the types of pre-processing done by the authors of the original datasets).

- A qualitative analysis of what HIVE-COTE v2.0 learned through the features of a few example datasets would tell the reader a lot more about why it performed so well.

**Strengths And Weaknesses:**

Strengths:

- Comparison of 30 EEG datasets from medical, BCI, and psychological domains, which is significant to the field.

- Researchers adhere to best practices in open science by fully sharing the data and code through the aeon-neuro repository.

- The comparative experiments utilize principled evaluation metrics (accuracy, balanced accuracy, AUROC, log loss), and evaluation methods utilize strong statistical inference (Wilcoxon signed-rank, Holm correction).

- The manuscript is well-organized, methodologically transparent, and communicates the findings clearly, including multiple tables that summarize the properties of the dataset and classifiers' performance.

- The research supports the use of TSML algorithms as baseline, reasonable, and sometimes state-of-the-art methods for EEG classification in low-data conditions.

- This work provides empirical evidence that ensemble-based TSML methods (HC2 in particular) generalize well across EEG problem types.

Weaknesses:

- The study adequately analyzes performance but fails to address neurophysiological implications and strengths of features, which is fundamentally important to the neuroscientific community.

- The authors’ report not performing any tuning across the classifiers, which could introduce a bias in results for deep models, especially.

- While HC2 is a very computationally expensive process (≈ 14 hours per dataset), the runtime is only discussed qualitatively, and there is no systematic analysis or scaling tests conducted to compare the runtime.

- Despite their recent popularity, this study is lacking benchmarking against pubs and models like EEGNet, DeepConvNet, or even transformer-based models with MedFormer, limiting its contemporary relevance to the EEG deep learning literature.

- Although the paper relies on datasets that have been preprocessed, utilizing different preprocessing pipelines/standards can introduce extracted bias across their datasets as well.

---

### Review · Reviewer_KBYA · 2025-11-02

**Recommendation:** 2
**Confidence:** 2

**Summary Of Contributions:**

The paper presents a large-scale empirical benchmark evaluating general-purpose time series machine learning (TSML) algorithms for electroencephalogram (EEG) classification. The authors introduce a curated EEG Classification Archive comprising 30 open-source datasets across medical, psychological, and brain-computer interface (BCI) domains, each with standardised preprocessing and train/test splits. The study compares eight classifiers, including HIVE-COTE v2.0 (HC2), MultiROCKET-Hydra, CNN, InceptionTime, and Riemannian geometry-based classifiers, using consistent statistical evaluation (Wilcoxon signed-rank test with Holm correction). The results show that HC2 achieves the highest accuracy and robustness across datasets, while MultiROCKET-Hydra offers a competitive trade-off between speed and performance. This work provides a reproducible foundation for benchmarking EEG classification methods and identifies promising TSML baselines for future studies.

**Strengths:**

This work contributes a robust, reproducible benchmark that fills a long-standing gap in EEG research, where models are often evaluated on small, non-standard datasets. The use of general-purpose TSC algorithms highlights their potential transferability beyond traditional domains. The quality of empirical evidence and breadth of datasets make the paper highly relevant to researchers working in data-centric ML, time-series analysis, and neuroinformatics. The analysis of HIVE-COTE’s internal diversity also reflects an admirable level of methodological transparency.

**Audience:**

Yes

**Broader Impact Concerns:**

No ethical or societal risks are evident. The datasets used are anonymized and publicly available, and the study does not engage in diagnostic claims. The authors may consider a brief note on responsible dataset use and data-sharing ethics, given the biomedical context.

**Claims And Evidence:**

The paper’s claims seem mostly supported by evidence.

**Datasets And Benchmarks:**

The dataset component is decent. Each dataset is described with acquisition parameters, train/test splits, and ethical context. Code and data availability on GitHub ensure reproducibility. The work meets the data documentation, hosting, and maintenance standards expected for benchmark papers.

**Extended Submissions:**

The submission appears original and not an extended version of a prior publication.

**Limitations:**

* The study’s contribution is empirical rather than conceptual.

* Results are limited by dataset size and computational constraints, which may underrepresent deep learning models’ potential.

* The benchmark assumes preprocessed, labelled EEG signals, limiting applicability to raw, uncleaned clinical data.

* HC2’s ensemble design trades interpretability and efficiency for accuracy, which may hinder deployment in real-time systems.

**Requested Changes:**

* Strengthen the methodological contribution by integrating domain-specific EEG properties (e.g., spatial–temporal correlations, Riemannian embeddings, or attention-based channel modelling).

* Include comparisons with modern EEG architectures (EEGNet, DeepConvNet, transformer-based models) to justify claims about deep learning limitations.

* Discuss potential runtime optimisation of HC2 or its components (e.g., parallelism, pruning).

* Provide error analysis or case studies for datasets where TSML fails.

* Condense Table 4 or move the extended results to an appendix for readability.

**Strengths And Weaknesses:**

Strengths:

* Comprehensive benchmark covering 30 EEG datasets, significantly expanding prior archives (e.g., MTSC).

* Rigorous experimental design with consistent preprocessing, fair comparison, and statistical validation of results.

* Strong focus on reproducibility, including open-source datasets and code through GitHub.

* Valuable diagnostic insights into HC2’s ensemble components (e.g., DrCIF, STC), improving interpretability.

* The manuscript is well written, logically structured, and easy to follow.

Weaknesses:

* no new theoretical contribution or insights beyond applying existing TSML frameworks.

*  CNN and InceptionTime are not state-of-the-art for EEG (e.g., no EEGNet, transformer-based, or hybrid models).

* Missing spatial–temporal modelling, EEG’s inter-channel correlations and spatial topologies are not incorporated into TSML methods.

* HC2’s computational cost (≈14h median per dataset) limits practical scalability; no concrete acceleration strategy is proposed.

* Statistical analysis could have been complemented by feature-level or frequency-domain interpretability.